# STAT3 and STAT5 Activation in Solid Cancers

**DOI:** 10.3390/cancers11101428

**Published:** 2019-09-25

**Authors:** Sebastian Igelmann, Heidi A. Neubauer, Gerardo Ferbeyre

**Affiliations:** 1Department of Biochemistry and Molecular Medicine, Université de Montréal, C.P. 6128, Succ. Centre-Ville, Montréal, QC H3C 3J7, Canada; sebastian.igelmann@umontreal.ca; 2CRCHUM, 900 Saint-Denis St, Montréal, QC H2X 0A9, Canada; 3Institute of Animal Breeding and Genetics, University of Veterinary Medicine Vienna, Vienna 1210, Austria; Heidi.Neubauer@vetmeduni.ac.at

**Keywords:** solid cancers, cell cycle, apoptosis, inflammation, mitochondria, stemness, tumor suppression

## Abstract

The Signal Transducer and Activator of Transcription (STAT)3 and 5 proteins are activated by many cytokine receptors to regulate specific gene expression and mitochondrial functions. Their role in cancer is largely context-dependent as they can both act as oncogenes and tumor suppressors. We review here the role of STAT3/5 activation in solid cancers and summarize their association with survival in cancer patients. The molecular mechanisms that underpin the oncogenic activity of STAT3/5 signaling include the regulation of genes that control cell cycle and cell death. However, recent advances also highlight the critical role of STAT3/5 target genes mediating inflammation and stemness. In addition, STAT3 mitochondrial functions are required for transformation. On the other hand, several tumor suppressor pathways act on or are activated by STAT3/5 signaling, including tyrosine phosphatases, the sumo ligase Protein Inhibitor of Activated STAT3 (PIAS3), the E3 ubiquitin ligase TATA Element Modulatory Factor/Androgen Receptor-Coactivator of 160 kDa (TMF/ARA160), the miRNAs miR-124 and miR-1181, the Protein of alternative reading frame 19 (p19ARF)/p53 pathway and the Suppressor of Cytokine Signaling 1 and 3 (SOCS1/3) proteins. Cancer mutations and epigenetic alterations may alter the balance between pro-oncogenic and tumor suppressor activities associated with STAT3/5 signaling, explaining their context-dependent association with tumor progression both in human cancers and animal models.

## 1. Introduction

Activation of Signal Transducer and Activator of Transcription (STAT) proteins has been linked to many human cancers. STATs were initially discovered as latent cytosolic transcription factors that are phosphorylated by the Janus Kinase (JAK) family upon stimulation of membrane-associated cytokine and growth factor receptors. Phosphorylation triggers STAT dimerization and translocation to the nucleus to bind specific promoters and regulate transcription [1]. Here, we review the role of STAT family members STAT3 and STAT5 in solid human malignancies, as well as the mechanisms that may explain their association with either worse or better prognosis. 

## 2. STAT3 and STAT5 in Solid Cancers

The discovery of cancer genes has been propelled by genetic analyses and more recently by next generation DNA sequencing technologies. Combined, these studies have identified 127 significantly mutated cancer genes that cover diverse signaling pathways [2]. Mutations acting as drivers in cancer are positively selected during tumor growth and constitute solid proof of the involvement of a particular gene as a driver in the disease. Mutations in STAT3 and STAT5 have been reported in patients with solid cancers, but unlike hyperactivation of the JAK/STAT pathway, STAT3/5 mutations in cancer are relatively infrequent and occur mostly in hematological malignancies. 

An overview of reported STAT3/5 mutations in solid cancers is illustrated in Figure 1, based on data collected from the Catalogue of Somatic Mutations in Cancer (COSMIC) database. Mutations in *STAT3* are more prevalent than mutations in *STAT5A* or *STAT5B* genes. Noticeably, gastrointestinal cancers have the highest rates of STAT3/5 mutations compared with other solid cancers (Figure 1). Missense mutations tend to cluster within the SH2 domain, where gain-of-function mutations were previously characterized [3,4], as well as within the DNA binding domain and to an extent the N-terminal domain (Figure 1A). Interestingly, the *STAT3* Tyrosine 640 into Phenylalanine (Y640F) hotspot gain-of-function mutation reported in various lymphoid malignancies has also been detected in patients with liver cancer (Figure 1A). Nonsense and frameshift mutations are less frequent and more disperse, likely representing loss-of-function events (Figure 1B). Notably, a hotspot frameshift mutation at position Q368 within the DNA binding domain of STAT5B has been reported in 24 patients with various types of carcinoma; this frameshift generates a stop codon shortly after the mutation and is therefore likely to be loss-of-function, although characterization of this mutation has not been performed.

As opposed to mutation rates, STAT3/5 activation is very frequent in human cancers, perhaps reflecting increased cytokine signaling or mutations in cytokine receptors or negative regulators. STAT3/5 activation can be detected using antibodies that measure total levels or activation marks in STAT3/5 proteins (e.g. tyrosine phosphorylation). A better assessment of STAT3/5 activation can be obtained by measuring downstream signaling targets (i.e., mRNA levels of STAT3 [5] and STAT5 [6] target genes). A recent metanalysis of 63 different studies concluded that STAT3 protein overexpression was significantly associated with a worse 3-year overall survival (OS) (OR = 2.06, 95% CI = 1.57 to 2.71, *p* < 0.00001) and 5-year OS (OR = 2.00, 95% CI = 1.53 to 2.63, *p* < 0.00001) in patients with solid tumors [7]. Elevated STAT3 expression was associated with poor prognosis in gastric cancer, lung cancer, gliomas, hepatic cancer, osteosarcoma, prostate cancer and pancreatic cancer. However, high STAT3 protein expression levels predicted a better prognosis for breast cancer [7]. This study mixed data of both STAT3 and phospho-STAT3 (p-STAT3) expression limiting its ability to associate pathway activation to prognosis. Here, we summarize the data linking activation of STAT3/5 to overall survival in several major human solid cancers identifying the biomarkers used in each study (Table 1). Taken together, the results clearly show that *STAT3* and *STAT5* are important cancer genes despite their relatively low mutation frequency.

STAT3 activation is clearly a factor linked to bad prognosis in patients with lung cancer, liver cancer, renal cell carcinoma (RCC) and gliomas. In other tumors, the association is not significant. In solid tumors, STAT3 activation is more frequent than STAT5 activation although no explanation for this difference was proposed. In prostate cancer, both STAT3 and STAT5 have been associated with castration-resistant disease and proposed as therapeutic targets [8,9]. In colon cancer, the association between p-STAT3 and survival varies according to the study, but a high p-STAT3/p-STAT5 ratio indicates bad prognosis [10]. Also in breast cancer, p-STAT5 levels are clearly associated with better prognosis [11]. In liver cancer, STAT5 has ambivalent functions that were recently reviewed by Moriggl and colleagues [12]. Understanding mechanistically how STAT3/5 promote transformation and tumor suppression is important for the eventual design of new treatments. Also, survival data is highly influenced by the response of patients to their treatment and may not always reflect all mechanistic links between STAT3/5 activity and tumor biology. Of note, the effect of any gene is conditioned by the genetic context of gene action. Some genes can clearly exert a tumor suppressor effect in the initial stages of carcinogenesis that is lost when cancer mutations or epigenetic changes inactivate key effectors of these tumor suppressor pathways [13]. Human studies are usually limited to late stage tumors because it is easier to collect samples at that point. Studies in model systems, including primary cells, organoids and mouse models are thus required for a full understanding of how cancer genes work specifically at early stages in tumorigenesis. 

## 3. Mechanisms of Transformation by STAT3/5 Proteins in Solid Cancers

STAT3 and STAT5 promote tumor progression by regulating the expression of cell cycle, survival and pro-inflammatory genes. In addition, they control mitochondrial functions, metabolism and stemness, as discussed below (Figure 2).

### 3.1. Cell Cycle and Apoptosis

As transcription factors, STAT3 and STAT5 regulate many genes required for cell cycle progression and cell survival. A major target of the transcriptional control of the mammalian cell cycle is cyclin D. STAT3 regulates cyclin D expression in a complex with CD44 and the acetyltransferase p300. The latter acetylates STAT3 promoting its dimerization, nuclear translocation and binding to the cyclin D promoter [25]. Other cell cycle and survival genes regulated by STAT3 include *c-MYC (myc proto-oncogene)*, B-cell lymphoma 2 (*BCL2)*, *BCL2L1*/BCL-XL (B-cell lymphoma-extra large), *MCL1* (Myeloid Cell Leukemia Sequence 1) and *BIRC5*/survivin [26]. Recent studies combined ChIPSeq with whole transcriptome profiling in ABC DLBCL (activated B cell-like diffuse large B cell lymphoma) cell lines and revealed that STAT3 activates genes in the Phosphoinositide 3-Kinase (PI3K)/AKT/Mammalian Target of Rapamycin (mTOR) pathway, the Nuclear Factor Kappa-Light-Chain Enhancer of Activated B-Cells (NF-κB) pathway and the cell cycle regulation pathway, while repressing type I interferon signaling genes [27]. STAT5 also regulates the expression of cell cycle and cell survival genes [13] including *AKT1* [28], which encodes a pro-survival kinase. 

### 3.2. Inflammation and Innate Immunity

Although the induction of cell proliferation and cell survival genes by STAT3/5 proteins contribute to their pro-cancer activity, in basal-like breast cancers the major genes associated with STAT3 activation control inflammation and the immune response [29]. Of note, inflammation is initially an adaptive response to pathological insults such as oncogenic stimuli, and it therefore exerts a tumor suppressive function. However, dysregulated inflammation in the long term provides a substrate for tumorigenesis [30]. STAT3 alone or in cooperation with NF-κB regulates the expression of many pro-inflammatory genes [31,32,33]. Starved tumor cells activate NF-κB and STAT3 via endoplasmic reticulum (ER) stress and secrete cytokines that stimulate tumor survival and clonogenic capacity [34]. The coactivation of these two transcription factors amplifies pro-inflammatory gene expression driving cancer-associated inflammation [35]. Of interest, the STAT3-NF-κB complex can repress the expression of DNA Damage Inducible Transcript 3 (DDIT3), an inhibitor of CCAAT Enhancer Binding Protein Beta (CEBPβ), another pro-inflammatory transcription factor [36]. 

Pharmacological agents that limit inflammation have been proposed for cancer prevention [37]. The use of metformin, a drug widely used to control diabetes, has been associated with a dramatic reduction in cancer incidence in many tissues [38]. Although the primary site of action of this drug is in mitochondria, a consequence of its effects is a potent reduction in the activation of NF-κB and STAT3, suggesting that the promising anticancer actions of metformin are related to its ability to curtail pro-inflammatory gene expression [39,40]. In contrast to STAT3, STAT5B inhibits NF-κB activity in the kidney fibroblast cell line COS by competing with coactivators of transcription [41], while it stimulates NF-κB in leukemia cells [42]. These results suggest the involvement of different regulatory mechanisms of STAT5 in hematopoietic cancers compared with solid cancers.

### 3.3. Mitochondria

In addition to their canonical roles in inflammation and immunity, STAT3 and STAT5 have been shown to localize to mitochondria. The mitochondrial localization of STAT3 is required for its ability to support malignant transformation in murine embryonic fibroblasts and breast cancer cells [43,44,45,46], and mito-STAT3 regulates mitochondrial metabolism and mitochondrial gene expression [45,47,48,49,50,51]. Several reports have suggested that STAT3 can be imported to mitochondria after phosphorylation on S727 [44,45] or upon acetylation [52,53]. Other studies have revealed that STAT3 mitochondrial translocation is mediated by interactions with Heat Shock Protein 22 (HSP22), Gene Associated with Retinoic and Interferon-Induced Mortality 19 (GRIM-19) or Translocase of Outer Mitochondrial Membrane 20 (TOM20) [54,55,56]. The mRNAs coding for some mitochondrial proteins are translated close to or in physical interaction with the import complex TOM [57,58]. The structural motifs mediating those interactions are located in the 3′ and 5′ UTRs of the mRNAs [59,60] and it will be interesting to investigate whether the mRNA of STAT3 also possesses RNA localization signals (zip codes) to localize in close proximity to mitochondria. 

Whereas the role of mitochondrial STAT3 has been extensively studied, the role of STAT5 in mitochondria is less clear. The import of STAT5 to mitochondria is regulated by cytokines [43]. Once imported into the mitochondria, STAT5 binds the D-loop of mitochondrial DNA, although no increase in transcription of mitochondrial genes was observed [61]. Mito-STAT5 is also able to interact with the Pyruvate Dehydrogenase Complex (PDC) and was shown to regulate metabolism towards glycolysis, as observed in cells treated with cytokines [43,61]. In the same line, STAT3 was also shown to interact with the PDC in mitochondria [53].

### 3.4. Reprogramming and Stemness

The role of STAT3 in stem cell biology was initially recognized due to the requirement for the cytokine LIF to maintain pluripotency in cultures of mouse embryonic stem (ES) cells. STAT3 activation mediates the induction or repression of several genes in mouse ES cells including the pluripotency factors *Oct4*, *Klf4*, *Tfcp2l1* and polycomb proteins [62,63,64]. Many pluripotency factors, such as Homeobox Protein NANOG, are short-lived proteins. STAT3 controls protein stability by inducing the expression of the deubiquitinase Ubiquitin Specific Peptidase 21 (USP21), stabilizing NANOG in mouse ES cells. Induction of ES cell differentiation promotes the Extracellular Signal-Regulated Kinase (ERK)-dependent phosphorylation of USP21 and its dissociation from NANOG, leading to NANOG degradation [65]. STAT3 also plays a role in the reprogramming of somatic cells into induced pluripotent stem (IPS) cells [66] and it has been suggested that its effects depend on the demethylation of pluripotency factor promoters [67]. STAT3 also activates mitochondrial DNA transcription, promoting oxidative phosphorylation during maintenance and induction of pluripotency [68]. It is thus likely that the ability of STAT3 to stimulate stemness also plays a role in its oncogenic activity. 

In many tumors, a subpopulation of cells possess a higher malignant capacity. These so-called tumor-initiating cells are suspected to regenerate the tumor after cancer chemotherapy and express many genes commonly expressed in ES cells [69]. It has been shown that STAT3 is required for the formation of tumor spheres and the viability of the cancer stem cell pool in many different tumors [39,40,70,71,72,73,74,75,76,77,78,79,80,81,82,83]. At least in breast cancer, a critical mechanism stimulated by STAT3 to regulate stemness involves genes in fatty acid oxidation [78,79] and the ability of STAT3 to adjust the levels of reactive oxygen species (ROS) produced in mitochondria [79]. In colorectal cancer cells, STAT3 forms a complex with the stem cell marker CD44 and the p300 acetyltransferase. Acetylation of STAT3 by this complex allows dimerization, nuclear translocation and binding to the promoters of genes required for stemness such as *c-MYC* and *TWIST1* [84]. 

The role of STAT5 in promoting cancer stemness does not affect many cell types and is mostly confined to hematopoietic cancers [85]. However, Nevalainen and colleagues reported that STAT5B induces stem cell properties in prostate cancer cells [86] in line with the increase in nuclear STAT5A/B observed in these tumors in correlation with bad prognosis [9]. Furthermore, transgenic mice with increased expression of prolactin in prostate epithelial cells displayed increases in the basal/stem cell compartment in association with activation of STAT5. This enrichment of stem cells was partially reversed by depletion of *Stat5a/b* [87]. The pro-stem cell oncogenic effect of STAT5 in the prostate contrasts with its effects in the mammary gland where STAT5 induces cell differentiation [88]. The ETS transcription factor Elf5 (E74-like factor 5) is a target of the prolactin-STAT5 axis and promotes mammary cell differentiation [89,90,91], supporting the tumor suppressive role of STAT5 in the mammary gland. 

## 4. Tumor Suppressor Functions and Negative Regulation of STAT3/5 Signaling

The oncogenic activity of JAK/STAT signaling is controlled by several molecular barriers that limit the activation of this pathway. They include tyrosine phosphatases, E3 SUMO ligases of the Protein Inhibitor of Activated STAT3 (PIAS) family, E3 ubiquitin ligases and miRNAs. In addition, oncogenic STAT3/5 signaling can activate fail-safe tumor suppressors such as protein of alternative reading frame 19 (p19ARF), Suppressor of Cytokine Signaling 1 (SOCS1) and p53 that trigger apoptosis, ferroptosis and/or senescence in potentially malignant cells (Figure 3). Understanding these different responses to STAT signaling in cancer is important to further distinguish tumors that would benefit from STAT3 or STAT5 inhibitors and those that would not. 

### 4.1. Tyrosine Phosphatases

Activation of STAT3 and STAT5 in tumors is often associated with tyrosine phosphorylation, a modification that can be reverted by several protein tyrosine phosphatases such as PTPN2 (Tyrosine-protein phosphatase non-receptor type 2), PTPN9/MEG2 (Tyrosine-protein phosphatase non-receptor type 9), PTPN11/SHP2 (Tyrosine-protein phosphatase non-receptor type 11) [92,93], CD45 [94] and PTPN6/SHP1 (Tyrosine-protein phosphatase non-receptor type 6) [95]. However, little is known about a possible role of these phosphatases in STAT3 activation in solid tumors. In liver cancers, SHP1 is downregulated in cells with mesenchymal features, and restoring its levels both reduced STAT3 phosphorylation and reversed the mesenchymal phenotype of liver cancer cells [95]. SHP1 and SHP2 also target STAT5 [96,97] but the significance of this regulation in solid tumors remains to be investigated.

### 4.2. PIAS

The Protein Inhibitor of Activated STAT3 (PIAS3) inhibits STAT3 transcriptional activity. In gliomas, PIAS3 expression is reduced [98]. Mechanistically, SMAD6 promotes PIAS3 degradation, promoting glioma cell growth and stem cell properties [76]. The PIAS proteins have SUMO E3 ligase activity acting on multiple proteins, and so their effects cannot be solely attributed to STAT3 inhibition [99]. Of interest, PIAS3 can bind NF-κB promoting its SUMOylation and inhibiting its activity [100,101], potentially targeting the expression of many pro-inflammatory genes required for tumor progression. Also, PIAS3 binds the N-terminus of p53 and prevents the interaction with its negative regulator MDM2, leading to p53 stabilization [102].

### 4.3. E3 Ligases

The Golgi resident and BC-box protein TATA Element Modulatory Factor/Androgen Receptor-Coactivator of 160 kDa (TMF/ARA160) was reported as an E3 ligase that catalyzes STAT3 ubiquitination leading to its proteasome-dependent degradation in myogenic C2C12 cells. The level of TMF/ARA160 was found to be significantly decreased in glioblastoma multiforme tumors, in benign meningioma and in malignant anaplastic meningioma, where STAT3 is known to play an oncogenic role [103]. TMF/ARA160 can also bind and ubiquitinate RELA/NF-κB leading to its proteasome-dependent degradation and a decrease in the expression of inflammatory genes [104]. Furthermore, the ubiquitin ligase TNF receptor associated factor 6 (TRAF6) binds and ubiquitinates STAT3 inhibiting the expression of STAT3 target genes [105]. During oncogene-induced senescence, STAT3 is degraded by the proteasome but the E3 ligase responsible has not been identified [106]. Recent results revealed that the long non-coding RNA (lncRNA) PVT1 (long non-coding RNA encoded by the human *PVT1* gene) binds STAT3 and protects it from ubiquitin-dependent degradation in gastric cancer [107]. PVT1 is upregulated in multiple cancers predicting poor prognosis for overall survival [108,109,110].

### 4.4. MiRNAs

The miRNA miR-124 regulates STAT3 signaling by targeting the mRNAs of interleukin-6 receptor (IL6R) [111] and STAT3 [112,113]. Suppression of this miRNA increases STAT3 phosphorylation and induces transformation in immortalized mouse hepatocytes. Of interest, systemic delivery of miR-124 prevented tumor growth in diethylnitrosamine (DEN)-treated mice, and miR-124 levels were found to be reduced in human hepatocellular carcinomas (HCC) [111]. In gliomas, miR-124 is poorly expressed but upregulation of its expression in glioma cancer stem cells inhibited the STAT3 pathway. In this model, STAT3 mediates immunosuppression, which was relieved upon systemic miR-124 delivery [114]. The circular RNA (circRNA) 100782 is upregulated in pancreatic cancer and its knockdown downregulates all miR-124 targets including IL6R and STAT3. This circRNA binds miR-124 suggesting that it may act as a miRNA sponge [115]. Furthermore, the miRNA miR-1181 also targets STAT3 and is downregulated in pancreatic cancer, predicting poorer overall survival. Overexpression of miR-1181 inhibited tumor formation and stem cell properties of pancreatic cancer cells [116].

### 4.5. The Suppressor of Cytokine Signaling SOCS

The members of the Suppressor of Cytokine Signaling (SOCS) family are major negative feedback regulators of JAK/STAT signaling and their expression is dysregulated in many human cancers [117,118,119]. These genes provide a barrier for cells with aberrant cytokine activation by inhibiting cytokine signaling [120]. In STAT3 driven cancers, SOCS3 seems to be the most important negative feedback regulator and mouse models of SOCS3 ablation show strong STAT3 activation [119,121,122,123,124]. On the other hand, in solid cancers where STAT5 plays a causal role such as liver and prostate cancer, in addition to SOCS3, SOCS1 is frequently inactivated and mouse models of SOCS1 ablation increase both liver and prostate tumorigenesis [125,126,127,128,129,130,131,132]. In addition to their role as JAK/STAT signaling barriers, SOCS1 and SOCS3 can bind p53 and activate tumor suppressor responses such as senescence and ferroptosis when their expression is induced by aberrant STAT5 signaling in primary cells [133,134,135,136,137,138]. In this way, SOCS1 and SOCS3 also act as fail-safe tumor suppressors in response to aberrant JAK/STAT signaling. So far, the SOCS1-p53-senescence axis has been demonstrated in primary fibroblasts and mammary epithelial cells [133,139,140,141]. This mechanism may explain the better prognosis of some solid cancers with high p-STAT5 [142,143,144] and the high frequency of SOCS1 inactivation in STAT5-driven cancers [125,126,127,128,129,130,131,132]. However, it is difficult to obtain evidence of a senescence tumor-suppression response by studying established tumors that have already circumvented this pathway. Senescence is particularly noticeable in premalignant lesions and benign tumors [40,106,145,146,147,148,149,150], and can be reactivated by cancer chemotherapy [151,152]. For this reason, evidence of STAT5-induced senescence in human cancers is not yet available and should be studied in samples from premalignant tumors or after chemotherapy.

The mechanisms that disable SOCS1 and SOCS3 in human cancers are often epigenetic, mediated either by miRNAs, promoter methylation or protein phosphorylation [127,128,130,131,137,153,154,155,156,157,158,159,160,161,162]. The SRC family of kinases (SFK) phosphorylate SOCS1 at Y80, interfering with p53-SOCS1 interactions. SFK inhibitors can reverse this effect and could be used to restore the SOCS1-p53 axis in tumors where these two proteins remain intact [162]. It is also possible to consider treatments that re-express SOCS1/3 in tumors. Indeed, in liver cancer *SOCS3* gene expression can be re-established by drugs that activate the Farnesoid X receptor (FXR) [163,164]. Gene therapy strategies are also under development to re-express SOCS1 or SOCS3 in tumors [165,166,167]. 

### 4.6. P19ARF-p53 Pathway

One of the first reports demonstrating that STAT3 can act as a tumor suppressor was shown in glioblastoma multiforme (GBM) [168] where a combination of low Phosphatase and tensin homolog (PTEN) expression and loss of STAT3 in astrocytes increased their tumorigenicity. This observation is in contrast to papers cited above on the requirement for STAT3 to maintain tumor stem cells in GBM [73,75,169]. This could be explained if STAT3 acts early in tumorigenesis as a tumor suppressor but gains oncogenic functions in the context of the cancer genome and epigenome. An interesting mechanism for the tumor suppressor role of STAT3 was recently described in the prostate where STAT3 induces the expression of p19ARF [170]. The latter is a tumor suppressor that activates p53 and inhibits ribosome biogenesis inducing cellular senescence and apoptosis [171,172,173,174]. Loss of STAT3 disrupts this STAT3-ARF-p53 axis and permits tumor progression [175]. STAT3 and other STATs can also induce p21 leading to cell cycle arrest or cellular senescence [176,177]. Further evidence for STAT3 as a tumor suppressor has been reported in lung [178], colon [179,180], thyroid [181], liver [182,183], skin [184], neck [185], nasopharynx, rectum [186], salivary gland [187] and breast cancers [188] but the mechanisms remain to be investigated. 

## 5. Concluding Remarks

Context-dependent activities of STAT3 and STAT5 in solid human cancers justify detailed molecular studies that will clarify the specific molecular mechanisms of action of these two cancer genes. The cancer genome and transcriptome are shaped and selected to favor cancer cell survival and proliferation. Although restoring mutated genes is technologically difficult, reprograming the transcriptome to restore tumor suppression may be feasible. Drugs acting on STAT3/5 and their regulators may restore the control of cell proliferation in cancer cells.

## Figures and Tables

**Figure 1 cancers-11-01428-f001:**
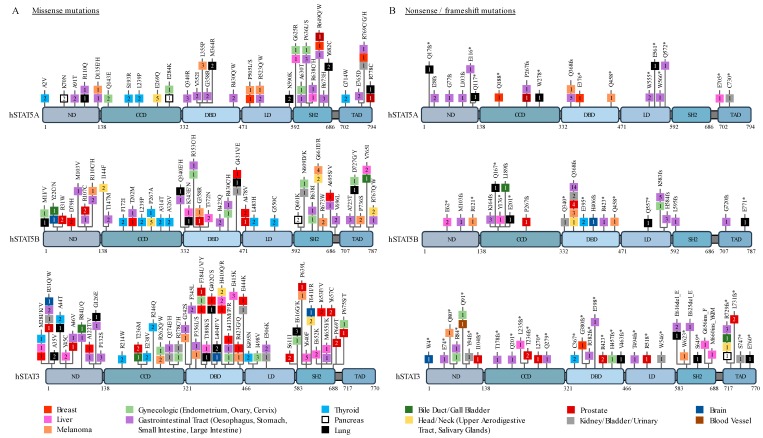
Map of somatic mutations detected in human Signal Transducer and Activator of Transcription (STAT)5A, STAT5B and STAT3 in patients with solid cancers. Individual missense mutations found in at least two patients (**A**), as well as all reported nonsense and frameshift mutations (**B**), are depicted. Numbers in each box represent the number of cases reported for each mutation. Data were mined from the Catalogue of Somatic Mutations In Cancer (COSMIC) database. ND, N-terminal domain; CCD, Coiled coil domain; DBD, DNA binding domain; LD, Linker domain; SH2, Src homology 2 domain; TAD, Transactivation domain.

**Figure 2 cancers-11-01428-f002:**
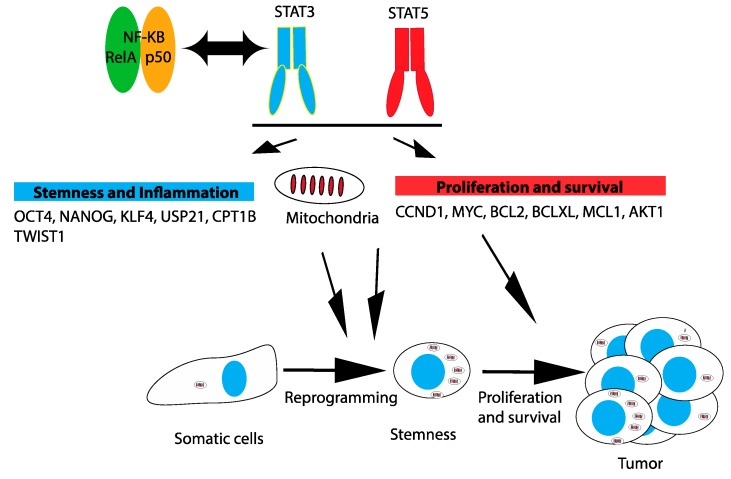
Mechanisms of tumorigenic activity of STAT3 and STAT5 signaling in solid tumors.

**Figure 3 cancers-11-01428-f003:**
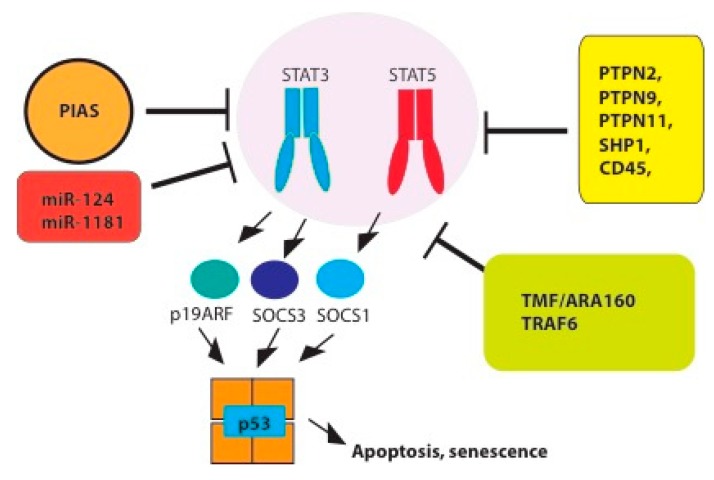
Tumor suppressor pathways acting on STAT3/5 activity (Protein Inhibitor of Activated STAT3 PIAS, miRNAs, E3 ligases, phosphatases) or activated by STAT3/5 transcriptional activity (Protein of alternative reading frame 19 (p19ARF) Suppressor of Cytokine Signaling 1 and 3 (SOCS1/3), p53). Abbreviations: (PTPN2 (Tyrosine-protein phosphatase non-receptor type 2), PTPN9/MEG2 (Tyrosine-protein phosphatase non-receptor type 9), PTPN11/SHP2 (Tyrosine-protein phosphatase non-receptor type 11), PTPN6/SHP1 (Tyrosine-protein phosphatase non-receptor type 6) and TNF receptor associated factor 6 (TRAF6)).

**Table 1 cancers-11-01428-t001:** STAT3/5 activity and overall survival in major human solid tumors.

Tumor Type	Biomarker/Type of Study	Overall Survival	Ref
NSCLC	High p-STAT3/Meta-analysis of 9 studies	Log HR 0.67, 95% CI: 0.57–0.77, *p* < 0.0001	[14]
NSCLC	High p-STAT3/Cox regression multivariate analysis	HR 2.45, 95% CI: 1.084–5.556, *p* = 0.031	[15]
Lung cancer	High p-STAT3/Meta-analysis of 13 studies	HR 1.23, 95% CI: 1.04–1.46, *p* = 0.02	[16]
Pancreatic cancer	High p-STAT3/Log-rank test	No association *p* > 0.05	[17]
Liver cancer (HCC)	High p-STAT3/Meta-analysis of 8 studies	HR 1.69, 95% CI: 1.07–2.31, *p* < 0.0001 3yr HR 1.67, 95% CI: 1.18–2.15, *p* < 0.0001 5yr	[18]
Breast cancer	High p-STAT3/Meta-analysis of 12 studies	No association *p* > 0.05	[19]
Breast cancer (ER+)	High p-STAT3/Log-rank test	No association *p* > 0.05	[20]
GBM	High p-S727-STAT3/Cox regression multivariate analysis	HR 1.797, 95% CI: 1.028–3.142, *p* = 0.040	[21]
RCC	High p-S727-STAT3/Cox regression multivariate analysis	HR 3.32, 95% CI: 1.26–8.71, *p* = 0.014 10yr	[22]
Colon cancer	High p-STAT3/p-STAT5 ratio/Cox regression multivariate analysis	HR 4.468, *p* = 0.043 5yr	[10]
Colon cancer	High p-STAT3/Log-rank test	Worse overall survival, *p* < 0.001	[23]
Colon cancer	High p-STAT3/Cox regression multivariate analysis	HR 1.61, 95% CI: 1.11–2.34, *p* = 0.015	[24]
Breast cancer	Low p-STAT5/Cox regression multivariate analysis	HR 2.49, 95% CI: 1.23–5.05, *p* = 0.012 5yr	[11]
Prostate cancer	High nuclear STAT5A/B/Cox regression multivariate analysis	HR 1.59, 95% CI: 1.04–2.44, *p* = 0.034	[9]

ER+, estrogen receptor-positive; HCC, hepatocellular carcinoma; GBM, glioblastoma multiforme; NSCLC, non-small-cell lung carcinoma; HR, hazard ratio; RCC, renal cell carcinoma.

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
