# Peer review of "STAT3 and STAT5 Activation in Solid Cancers"

_cancers, 2019, doi:10.3390/cancers11101428_

Round 1

Reviewer 1 Report

This review from Igelmann, Neubauer and Ferbeyre describes well the role of Stat3 and Stat5 in solid tumor, focusing first on cell processes regulated by oncogenic Stat3/5, then on molecular mechanisms by which Stat3/5 acts as tumor suppressors.

I have identified some points that need to be clarified:

-line 109: the official gene name of survivin is BIRC5 (Homo sapiens) or Birc5 (Mus musculus), also known as survivin40.

-line 116: the role of Stat3 in this context is not clear. By inducing inflammation, does Stat3 promotes or blocks tumor formation?

-line 124: why is metformin mentioned here? what is the link with the rest of the paragraph?

-line 152: Klf4 and Tfcp2l1 are known direct transcriptional targets in mouse ES cells, while regulation of Nanog via USP21 is clearly post-transcriptional. Better to clarify this point. Moreover it should be clearly stated whether human or mouse ES cells were used in such studies.

-line 164: the idea that a common gene program underpins both cancer and stemness is debatable and potentially misleading. Multiple stem cell types exist, such as Embryonic Stem cells, Hematopoietic Stem cells and Neural Stem cells and several studies failed at finding any common gene signature associated with "stemness" ( see https://science.sciencemag.org/content/302/5644/393.3 ).

Rather than a "universal signature" of stemness, there might be genes specifically expressed by some stem cells and not by others, so I would suggest to state that some genes are expressed by cancer cells and Embryonic Stem cells.

Moreover, given that stem cells and cancers cells often display high proliferation rate, the common gene program (if present) might simply represent a general proliferation gene program.

-line 186: it would be important to mention that Stat3 has also been reported as promoter of GBM stem cells (doi: 10.1002/stem.185.)

-line 189: please briefly explain what p19ARF is. Similarly, some other proteins are mentioned without an explanation of what their function is. 

-line 212: in what cancer does Stat5 regulate senescence and ferroptosis?

-line 235: better to write "in glioblastoma multiform tumors, in benign meningioma and in malignant anaplastic meningioma tumors" rather than "malignant brain tumors"

line 252: circRNA is the correct abbreviation for circular RNAs.

Author Response

This review from Igelmann, Neubauer and Ferbeyre describes well the role of Stat3 and Stat5 in solid tumor, focusing first on cell processes regulated by oncogenic Stat3/5, then on molecular mechanisms by which Stat3/5 acts as tumor suppressors.

I have identified some points that need to be clarified:

-line 109: the official gene name of survivin is BIRC5 (Homo sapiens) or Birc5 (Mus musculus), also known as survivin40.

We thank the reviewer for pointing this out. We have now included the gene name for surviving and also for BCL-XL (line 105).

-line 116: the role of Stat3 in this context is not clear. By inducing inflammation, does Stat3 promotes or blocks tumor formation?

The reviewer is right. We now write:

“Of note, inflammation is initially an adaptive response to pathological insults such as oncogenic stimuli, and it therefore exerts a tumor suppressive function. However, dysregulated inflammation in the long term provides a substrate for tumorigenesis.” (lines 117-121).

In addition, we have now cited a recent review summarizing both the oncogenic and tumor suppressive roles of inflammation: doi:10.1038/s41568-019-0123-y.

-line 124: why is metformin mentioned here? what is the link with the rest of the paragraph?

We have now moved the discussion on metformin to a new paragraph and added a better introduction to it as follows:

“Pharmacological agents that limit inflammation have been proposed for cancer prevention. The use of metformin, a drug widely used to control diabetes, has been associated with a dramatic reduction in cancer incidence in many tissues. Although the primary site of action of this drug is in mitochondria, a consequence of its effects is a potent reduction in the activation of NF-kB and STAT3, suggesting that the promising anticancer actions of metformin are related to its ability to curtail pro-inflammatory gene expression.” (lines 128-133).

-line 152: Klf4 and Tfcp2l1 are known direct transcriptional targets in mouse ES cells, while regulation of Nanog via USP21 is clearly post-transcriptional. Better to clarify this point. Moreover it should be clearly stated whether human or mouse ES cells were used in such studies.

We have added a reference to the paper by Ye et al. on Tfcp2l1 and have clarified that Nanog is not a direct target of the transcriptional activity of Stat3. We have also now indicated that mouse ES cells were used in studies linking Stat3 to ES cells (lines 160-166).

-line 164: the idea that a common gene program underpins both cancer and stemness is debatable and potentially misleading. Multiple stem cell types exist, such as Embryonic Stem cells, Hematopoietic Stem cells and Neural Stem cells and several studies failed at finding any common gene signature associated with "stemness" ( see https://science.sciencemag.org/content/302/5644/393.3 ).

Rather than a "universal signature" of stemness, there might be genes specifically expressed by some stem cells and not by others, so I would suggest to state that some genes are expressed by cancer cells and Embryonic Stem cells.

Moreover, given that stem cells and cancers cells often display high proliferation rate, the common gene program (if present) might simply represent a general proliferation gene program.

We agree and deleted the statement in question.

-line 186: it would be important to mention that Stat3 has also been reported as promoter of GBM stem cells (doi: 10.1002/stem.185.)

We added the reference to the section discussing Stat3 in cancer stem cells and the following comment to the section in tumor suppression:

“This observation is in contrast to papers cited above on the requirement for STAT3 to maintain tumor stem cells in GBM. This could be explained if STAT3 acts early in tumorigenesis as a tumor suppressor but gain oncogenic functions in the context of the cancer genome and epigenome.” (lines 285-288).

-line 189: please briefly explain what p19ARF is. Similarly, some other proteins are mentioned without an explanation of what their function is. 

For a description of p19ARF, we have added:

“The latter is a tumor suppressor that activates p53 and inhibits ribosome biogenesis inducing cellular senescence and apoptosis.” (line2 290-291).

-line 212: in what cancer does Stat5 regulate senescence and ferroptosis?

Stat5 regulation of senescence and ferroptosis has been observed in primary cells (fibroblasts and mammary epithelial cells). In human cancers, these pathways are inactivated so it is not possible to observe this phenomenon. To clarify, we have now included:

“In addition to their role as JAK/STAT signaling barriers, SOCS1 and SOCS3 can bind p53 and activate tumor suppressor responses such as senescence and ferroptosis when their expression is induced by aberrant STAT5 signaling in primary cells. In this way, SOCS1 and SOCS3 also act as fail-safe tumor suppressors in response to aberrant JAK/STAT signaling. So far, the SOCS1-p53-senescence axis has been demonstrated in primary fibroblasts and mammary epithelial cells. This mechanism may explain the better prognosis of some solid cancers with high p-STAT5 and the high frequency of SOCS1 inactivation in STAT5-driven cancers. However, it is difficult to obtain evidence of a senescence tumor-suppression response by studying established tumors that have already circumvented this pathway. Senescence is particularly noticeable in premalignant lesions and benign tumors and can be reactivated by cancer chemotherapy. For this reason, evidence of STAT5-induced senescence in human cancers is not yet available and should be studied in samples from premalignant tumors or after chemotherapy.” (lines 257-269).

-line 235: better to write "in glioblastoma multiform tumors, in benign meningioma and in malignant anaplastic meningioma tumors" rather than "malignant brain tumors"

We agree and have corrected the text (lines 225-226).

line 252: circRNA is the correct abbreviation for circular RNAs.

We have corrected the text (lines 243 and 245).

Reviewer 2 Report

This is a most useful overview summarizing literature on Stat3 and STAT5 activation in solid cancers. It not only focusses on their oncogenic role in cancer, it also discusses tumor suppressor activities associated with STAT3 and STAT5 in detail.

To summarize, the review article is well written and should be published in Cancers without further revision.

Author Response

We thank the reviewer for his/her time and positive comments.

Reviewer 3 Report

Good logical presentation. English editing should done.

Author Response

We thank the reviewer for his/her positive comment. The submitted version of our manuscript had been edited by a native English speaker and we feel that the English language and style of the manuscript was already of a high level, which was also the opinion of all other reviewers. However, in preparing this resubmission we have again carefully edited the manuscript for English language and style.

Reviewer 4 Report

This is a nice short overview of the role of STAT3/5 in solid cancers supported by an extensive list of references which will help the reader to find additional information on the tumor or STAT protein of interest.

I have one major comment and a few minor suggestions regarding the formatting of text and figures.

Major comment

The tumor suppressor section (lines 176 and further) mixes i) tumor suppressor mechanisms that are activated by STAT3/5 (and that oppose to their "classical" tumor promoting effects) and ii) mechanisms that suppress (negatively regulate) STAT3/5 activation. In the former case, high levels of STAT3/5 activation correlate good prognosis due to STAT-mediated activation of tumor suppressor pathways, e.g. STAT3/ARF/p53 pathway in prostate cancer or STAT5/SOCS1/p53 pathway in other tumors. In the latter case, the deficiency in these negative regulators leads to high STAT3/5 activation in tumors which correlates bad prognosis due to classical oncogenic activity of STAT3/5. A typical example is provided in the Tyrosine phosphatases paragraph: high STAT3 activation in liver cancer (which is of bad prognosis from Table 1) is due to low SHP1 expression as restoration of SHP1 expression reverses the mesenchymal (aggressive) phenotype. This reviewer understands that the paragraphs on E3 ligases and miRNAs also exclusively relate to various mechanisms that regulate STAT3 oncogenic activity (and not to any intrinsic tumor suppressor activity triggered by STAT3).  

The context is more complex for SOCS and PIAS3 proteins which exert dual functions. Related to their canonical function (JAK/STAT downregulation that should also be depicted in Figure 3 for SOCS), SOCS/PIAS3 inactivation promotes STAT hyper-activation and in turn tumorigenesis, which correlates the "classical" tumor-promoting action of STATs (line 209-211 and 224-226). Otherwise, SOCS1/3 and PIAS3 on their own activate p53 or other mechanisms, supporting a "non-classical" tumor-suppressor function of STAT3/5 pathways.

In summary, this reviewer was a bit confused about this section of the review and believes it may be suitable to better distinguish STAT-triggered tumor suppressor pathways (which for most are probably less/not known from the readers) from mechanisms that explain how STAT activity can be increased in tumors, which correlates with classical pro-tumor effect.  

Minor comments

Fig 1 is very busy and is difficult to read in its current form. We suggest to increase its size and to superimpose the two blocks of 3 panels instead of displaying them side by side.  

Line 65: for non-specialist readers it may be interesting to provide a list of classical STAT3 and STAT5 target genes (overlapping versus different) or to refer to papers in which this information can be found.

In Table 1, it may welcome to better display the biomarker analyzed (which is of primary interest for the readers) e.g. write it before the type of statistical analysis, or make an individual column. Also, as "ER+ BC" may be unclear for non-specialists, I suggest to write "breast cancer (ER+)" and state in the legend what ER+ means. Note that on line 120 "ER" is used for endoplasmic reticulum, so using the same abbreviation for two different meanings is confusing. As a general comment, many abbreviations are not defined (e.g. DDIT3, CEBP, GRIM19, TOM20, etc).

Title on line 97 should be moved to next page

Line 125-127: is there any hypothesis on the mechanisms underlying opposite, cell-specific effects of STAT5 on NFkB?

Line 132: it may be welcome to indicate in which types of cancers mito-STAT3 supports transformation (ref 38-41)

Line 139: can you precise what you mean by "RNA zip code" biologically speaking?

Line 172-175: the observation of Nevalainen regarding STAT5-induced stem properties in prostate cancer cells is further supported by in vivo observations involving mice overexpressing PRL in the prostate in which basal/progenitor cells are amplified at pre-neoplastic stages. The article by Boutillon et al in this special Issue of Cancers dissects the role of STAT5 in this process. Also, various groups (Ormandy, Wagner, Rui) have shown that the PRLR/STAT5/Elf5 axis promotes mammary cell differentiation at the expense of stem-like characteristics, which support the protective role of STAT5 signaling in breast cancer (Table 1).The authors may consider referring to these studies in this section.

Lines 201-202: have the mechanisms underlying the tumor suppressor function of STAT3 been elucidated in all the tissues cited here? Is it p21-mediated? Please clarify or provide a little more information.   

Author Response

This is a nice short overview of the role of STAT3/5 in solid cancers supported by an extensive list of references which will help the reader to find additional information on the tumor or STAT protein of interest.

I have one major comment and a few minor suggestions regarding the formatting of text and figures.

Major comment

The tumor suppressor section (lines 176 and further) mixes i) tumor suppressor mechanisms that are activated by STAT3/5 (and that oppose to their "classical" tumor promoting effects) and ii) mechanisms that suppress (negatively regulate) STAT3/5 activation. In the former case, high levels of STAT3/5 activation correlate good prognosis due to STAT-mediated activation of tumor suppressor pathways, e.g. STAT3/ARF/p53 pathway in prostate cancer or STAT5/SOCS1/p53 pathway in other tumors. In the latter case, the deficiency in these negative regulators leads to high STAT3/5 activation in tumors which correlates bad prognosis due to classical oncogenic activity of STAT3/5. A typical example is provided in the Tyrosine phosphatases paragraph: high STAT3 activation in liver cancer (which is of bad prognosis from Table 1) is due to low SHP1 expression as restoration of SHP1 expression reverses the mesenchymal (aggressive) phenotype. This reviewer understands that the paragraphs on E3 ligases and miRNAs also exclusively relate to various mechanisms that regulate STAT3 oncogenic activity (and not to any intrinsic tumor suppressor activity triggered by STAT3).  

The context is more complex for SOCS and PIAS3 proteins which exert dual functions. Related to their canonical function (JAK/STAT downregulation that should also be depicted in Figure 3 for SOCS), SOCS/PIAS3 inactivation promotes STAT hyper-activation and in turn tumorigenesis, which correlates the "classical" tumor-promoting action of STATs (line 209-211 and 224-226). Otherwise, SOCS1/3 and PIAS3 on their own activate p53 or other mechanisms, supporting a "non-classical" tumor-suppressor function of STAT3/5 pathways.

In summary, this reviewer was a bit confused about this section of the review and believes it may be suitable to better distinguish STAT-triggered tumor suppressor pathways (which for most are probably less/not known from the readers) from mechanisms that explain how STAT activity can be increased in tumors, which correlates with classical pro-tumor effect.  

We thank the reviewer for this important point and agree that it needs clarification. We have re-worded the section title to “Tumor suppressor functions and negative regulation of STAT3/5 signaling”. Furthermore, we have better introduced this section to define the different mechanisms:

“The oncogenic activity of JAK/STAT signaling is controlled by several molecular barriers that limit the activation of this pathway. They include tyrosine phosphatases, E3 SUMO ligases of the PIAS family, E3 ubiquitin ligases and miRNAs. In addition, oncogenic STAT3/5 signaling can activate fail-safe tumor suppressors such as p19ARF, SOCS1 and p53 that trigger apoptosis, ferroptosis and/or senescence in potentially malignant cells (Figure 3).” (lines 194-198).

Minor comments

Fig 1 is very busy and is difficult to read in its current form. We suggest to increase its size and to superimpose the two blocks of 3 panels instead of displaying them side by side.  

We agree with the reviewer that Figure 1 should be enlarged. The figure was intended to be displayed in landscape orientation and to take up one whole page. We have now corrected the orientation and increased the size to make it more legible. 

Line 65: for non-specialist readers it may be interesting to provide a list of classical STAT3 and STAT5 target genes (overlapping versus different) or to refer to papers in which this information can be found.

We have added two references that report lists of targets for STAT3 and STAT5 (line 61):

1) Carpenter, R.L.; Lo, H.W. STAT3 Target Genes Relevant to Human Cancers. Cancers (Basel) 2014, 6, 897-925, doi:10.3390/cancers6020897.

2) Basham, B.; Sathe, M.; Grein, J.; McClanahan, T.; D'Andrea, A.; Lees, E.; Rascle, A. In vivo identification of novel STAT5 target genes. Nucleic Acids Res 2008, 36, 3802-3818, doi:gkn271 [pii].

In Table 1, it may welcome to better display the biomarker analyzed (which is of primary interest for the readers) e.g. write it before the type of statistical analysis, or make an individual column. Also, as "ER+ BC" may be unclear for non-specialists, I suggest to write "breast cancer (ER+)" and state in the legend what ER+ means. Note that on line 120 "ER" is used for endoplasmic reticulum, so using the same abbreviation for two different meanings is confusing. As a general comment, many abbreviations are not defined (e.g. DDIT3, CEBP, GRIM19, TOM20, etc).

We have made the suggested corrections.

Title on line 97 should be moved to next page

We have now moved the title.

Line 125-127: is there any hypothesis on the mechanisms underlying opposite, cell-specific effects of STAT5 on NFkB?

In the paper cited, the authors concluded that the mechanism involved competition for common coactivators. We have now added these details to the text (line 134).

Line 132: it may be welcome to indicate in which types of cancers mito-STAT3 supports transformation (ref 38-41)

We have now included this information (line 140).

Line 139: can you precise what you mean by "RNA zip code" biologically speaking?

Zip codes are RNA localisation signals. We have now clarified the text: “the mRNA of STAT3 also possesses RNA localization signals (zip codes) to localize in close proximity to mitochondria.” (lines 149-150).

Line 172-175: the observation of Nevalainen regarding STAT5-induced stem properties in prostate cancer cells is further supported by in vivo observations involving mice overexpressing PRL in the prostate in which basal/progenitor cells are amplified at pre-neoplastic stages. The article by Boutillon et al in this special Issue of Cancers dissects the role of STAT5 in this process. Also, various groups (Ormandy, Wagner, Rui) have shown that the PRLR/STAT5/Elf5 axis promotes mammary cell differentiation at the expense of stem-like characteristics, which support the protective role of STAT5 signaling in breast cancer (Table 1).The authors may consider referring to these studies in this section.

We thank the reviewer for all of these very intriguing suggestions. We now write:

“Furthermore, transgenic mice with increased expression of prolactin in prostate epithelial cells displayed increases in the basal/stem cell compartment in association with activation of STAT5. This enrichment of stem cells was partially reversed by depletion of Stat5a/b. The pro-stem cell-oncogenic effect of STAT5 in the prostate contrasts with its effects in the mammary gland where STAT5 induces cell differentiation. The ETS transcription factor Elf5 (E74-like factor 5) is a target of the prolactin-STAT5 axis and promotes mammary cell differentiation, supporting the tumor suppressive role of STAT5 in the mammary gland.” (lines 185-192).

Lines 201-202: have the mechanisms underlying the tumor suppressor function of STAT3 been elucidated in all the tissues cited here? Is it p21-mediated? Please clarify or provide a little more information.  

The mechanisms of the reported tumor suppressor activity of STAT3 remain for the most part unknown. We have clarified that in the text (line2 295-296). 

Round 2

Reviewer 4 Report

The authors took into accounts all my suggestions, and in my opinion the review manuscript is now much improved and acceptable in its present form.